# The Role of Injectable Platelet-Rich Fibrin in Orthopedics: Where Do We Stand?

**DOI:** 10.3390/cimb47040239

**Published:** 2025-03-29

**Authors:** Fábio Ramos Costa, Sergio Augusto Lopes de Souza, Rubens Andrade Martins, Bruno Ramos Costa, Luyddy Pires, Alex Pontes de Macedo, Napoliane Santos, Stephany Cares Huber, Gabriel Silva Santos, André Kruel, Márcia Santos, José Fábio Lana

**Affiliations:** 1Department of Orthopedics, FC Sports Traumatology, Salvador 40296-210, BA, Brazil; 2Department of Radiology, Federal University of Rio de Janeiro (UFRJ), Rio de Janeiro 21941-916, RJ, Brazil; sergioalsouza@gmail.com; 3Medical School, Tiradentes University Center, Maceió 57038-000, AL, Brazil; rubensdeandrade@hotmail.com; 4Medical School, Zarns College, Salvador 41720-200, BA, Brazil; fabiocosta7113@gmail.com; 5Department of Orthopedics, Brazilian Institute of Regenerative Medicine (BIRM), Indaiatuba 13334-170, SP, Brazil; luyddypires@gmail.com (L.P.); alex_macedo@icloud.com (A.P.d.M.); dranapolianesantos@gmail.com (N.S.); stephany_huber@yahoo.com.br (S.C.H.); josefabiolana@gmail.com (J.F.L.); 6Regenerative Medicine, Orthoregen International Course, Indaiatuba 13334-170, SP, Brazil; kruel.andre@gmail.com; 7Nutritional Sciences, Metropolitan Union of Education and Culture, Salvador 42700-000, BA, Brazil; marcinha_mairi@hotmail.com; 8Medical School, Max Planck University Center (UniMAX), Indaiatuba 13343-060, SP, Brazil; 9Clinical Research, Anna Vitória Lana Institute (IAVL), Indaiatuba 13334-170, SP, Brazil; 10Medical School, Jaguariúna University Center (UniFAJ), Jaguariúna 13911-094, SP, Brazil

**Keywords:** platelet-rich fibrin, orthopedics, tissue regeneration, anti-inflammatory therapy, regenerative medicine

## Abstract

Injectable Platelet-Rich Fibrin (i-PRF) has emerged as a promising tool in regenerative medicine, particularly in orthopedics, due to its unique biological properties and ease of preparation. i-PRF is an autologous platelet concentrate derived through a simple, anticoagulant-free centrifugation process, resulting in a liquid matrix enriched with fibrin, leukocytes, and growth factors. These components promote tissue regeneration, angiogenesis, and anti-inflammatory responses, making i-PRF suitable for bone and cartilage repair as well as drug delivery systems. This review discusses the history, biological mechanisms, and clinical applications of i-PRF in orthopedics, highlighting its potential advantages over traditional platelet-rich plasma (PRP). Furthermore, we address the challenges and limitations of i-PRF, including drug stability, release control, and bioactive interactions, underscoring the need for further research to optimize its therapeutic efficacy.

## 1. Introduction

Platelet-rich fibrin (PRF) is a second-generation platelet concentrate that was first introduced in the field of oral and maxillofacial surgery by Choukroun and colleagues [1]. PRF was initially used in oral and maxillofacial surgical procedures in 2001 by Choukroun et al. due to its simplicity, cost-effectiveness, and ease of handling [2]. This biological product was conceived as a promising alternative to existing bone grafts and platelet-rich plasma at the time [3].

This autologous biomaterial contains a dense fibrin matrix, along with leukocytes and a wide range of healing proteins [4]. Unlike other platelet preparations, such as PRP, i-PRF is obtained through a single centrifugation process that minimizes blood manipulation and eliminates the use of anticoagulants [4]. Coagulation is an essential step in healing, and the inclusion of anticoagulants during the preparation process of i-PRF hinders the maximization of its regenerative potential [4].

Due to the natural tendency of platelet-rich fibrin to coagulate during centrifugation, centrifugation speeds were reduced to produce a liquid, non-coagulated version of PRF, which later became known as i-PRF or injectable PRF [4]. This liquid form of PRF contained liquid fibrinogen and thrombin that had not yet been converted into fibrin, resulting in improved wound healing due to its clotting ability [5].

In 2015, i-PRF was developed and initially investigated using a very short and slow centrifugation protocol at 700 rpm (60 G) for 3–4 min with plastic tubes [6]. Since then, various basic research studies have demonstrated the regenerative potential of i-PRF compared to PRP.

Using the technique mentioned above, Miron et al. observed a 2.07-fold increase in platelet concentrations, as well as a 23% increase in leukocytes. Furthermore, it was revealed that horizontal centrifugation of PRF, as opposed to standard fixed-angle devices, resulted in a fourfold increase in cell concentration [7].

Currently, one of the most commonly used protocols involves collecting 10 mL of venous blood in a glass or plastic tube (dry tube) without anticoagulants and performing centrifugation for 5 to 8 min at 60 G (Figure 1). This process results in an upper liquid layer of i-PRF that concentrates 2 to 3 times more platelets than whole blood [7,8,9].

## 2. Biological Properties

i-PRF consists of a fibrin matrix containing a high concentration of platelets, leukocytes, and a variety of growth factors. The growth factors present in i-PRF include platelet-derived growth factor (PDGF), transforming growth factor beta (TGF-β), fibroblast growth factor (FGF), and vascular endothelial growth factor (VEGF) [4]. These growth factors play a crucial role in tissue regeneration by stimulating cell proliferation, differentiation, and migration, as well as promoting angiogenesis [10]. Additionally, the fibrin matrix of i-PRF mimics the natural extracellular matrix and provides a bioactive scaffold for tissue regeneration [4].

Regarding its ability to stimulate angiogenesis, i-PRF demonstrates remarkable potential. Its fibrin matrix activates various mechanisms that promote the formation of new blood vessels, including the release of angiogenic growth factors such as VEGF [11]. This ability to stimulate angiogenesis is crucial for tissue regeneration and healing, as an adequate blood supply is essential to deliver the oxygen and nutrients required for repair processes [12]. The formation of new blood vessels facilitates the delivery of essential elements that support cellular activities, such as the proliferation, migration, and differentiation of cells involved in tissue repair and regeneration [12]. This process of neovascularization ensures an adequate exchange of metabolites and gases, which is vital for the successful progression of the healing cascade [13].

The anti-inflammatory properties of i-PRF are mediated by its ability to modulate the inflammatory response and promote the resolution of inflammation [14]. Several studies have demonstrated that i-PRF can reduce the expression of pro-inflammatory cytokines, such as interleukin-1 beta (IL-1β) and tumor necrosis factor-alpha (TNF-α), while increasing the expression of anti-inflammatory cytokines, such as interleukin-10 (IL-10) [14,15,16]. This modulation of the inflammatory response is essential to prevent tissue damage and allow the orderly progression of the healing process.

Another notable biological property of i-PRF is its ability to promote the polarization of macrophages from the pro-inflammatory M1 phenotype to the anti-inflammatory and regenerative M2 phenotype [14]. This phenotypic shift in macrophages plays a crucial role in modulating the inflammatory response and promoting tissue healing and regeneration, as it prevents excessive inflammation that could hinder the healing process [17].

In a study conducted by Nasirzade et al., researchers observed that exposing primary murine macrophages and a human macrophage cell line to saliva and lipopolysaccharides, along with PRF lysates, resulted in a significant reduction in the expression of pro-inflammatory M1 marker genes, such as IL-1β and IL-6 [14]. In parallel, the PRF-conditioned medium increased the expression of tissue resolution markers. These findings led the researchers to conclude that PRF possesses potent anti-inflammatory activity and is capable of altering macrophage polarization from a pro-inflammatory M1 phenotype to an anti-inflammatory and regenerative M2 phenotype.

### 2.1. Antibacterial Properties of i-PRF

i-PRF exhibits antibacterial properties that help prevent infections at the application site. Several studies have demonstrated that i-PRF can inhibit the growth of bacteria commonly associated with infections, such as Escherichia coli, Staphylococcus aureus, and Pseudomonas aeruginosa [18]. The antimicrobial capacity of i-PRF can be attributed to the presence of antibacterial factors, such as lysozyme, and the formation of a physical barrier that prevents bacterial proliferation [19]. This characteristic adds a significant benefit to the use of i-PRF in clinical applications, reducing the risk of infectious complications and enhancing its safety. The antimicrobial properties of i-PRF can also be attributed to the presence of leukocytes, particularly neutrophils and lymphocytes, in its composition. These cellular elements act as immune sentinels, contributing to the elimination of invading pathogens and preventing the progression of potential infections at the i-PRF application site [20].

### 2.2. i-PRF as a Pharmacological Carrier

i-PRF has also demonstrated its potential as a drug delivery system [21]. Due to its unique fibrin matrix structure, i-PRF can serve as a carrier and release platform for various therapeutic agents, such as antibiotics, anti-inflammatory drugs, and growth factors. This capability enables localized and sustained delivery of these agents to the target tissue, enhancing their therapeutic efficacy while minimizing systemic side effects [21,22,23]. The ability of i-PRF to function as a drug delivery system is a significant advantage, as it allows for the incorporation of tailored treatments specific to the patient’s needs and a particular injury or condition being addressed.

A key advantage of i-PRF compared to PRP is its gradual and prolonged release of growth factors over time. While PRP releases growth factors within minutes to hours after application, i-PRF provides a much longer release, typically extended over a period of 14 days [24]. This is an advantage that makes i-PRF comparable to PRP, as even with a lower platelet concentration, it can release growth factors over a longer period, thereby promoting better regeneration.

The fibrin matrix of i-PRF functions as a scaffold that helps retain large and small molecules at the application site for an extended period. These molecules are considered potential tools capable of targeting specific sites to enhance the regeneration of bone and cartilage tissue [4].

Drawing an analogy between construction and regenerative medicine, one can imagine a worker laying bricks in a building. If this worker was to be surrounded by a network of reinforcement bars (scaffolding), they would have more difficulty moving freely within that area [25]. Consequently, they would end up performing their work gradually and sustainably, rather than in a rapid flow. This is due to the presence of the support structure (scaffolding), which metaphorically represents the fibrin network.

In the biological context of regenerative medicine, the worker would represent the therapeutic agent, while the reinforcement bars would symbolize the fibrin matrix, enabling the gradual and sustained release of the agent as a result of this scaffolding structure. Just as the worker’s movements are restricted by the reinforcement bars, the fibrin matrix of i-PRF acts as a scaffold that helps retain large and small molecules at the application site for an extended period, resulting in the gradual and sustained release of therapeutic agents as a true and efficient drug delivery system [25].

Despite the great potential of i-PRF as a drug delivery system, several important issues need to be addressed. Among them is the stability of the drug within the fibrin matrix, which can be influenced by the biochemical properties of i-PRF. Furthermore, the precise control of the release rate of therapeutic agents presents a challenge, particularly given the complexity of interactions between the drug and the biological microenvironment [26]. Finally, potential interactions between the drug and the bioactive components of i-PRF, such as cytokines, growth factors, and cells, may impact both the drug’s efficacy and the properties of i-PRF. These aspects highlight the need for further studies to optimize this application and ensure its clinical safety and efficacy.

### 2.3. i-PRF in Bone Regeneration

The use of i-PRF has shown promising results in the field of bone regeneration. In vitro studies have demonstrated that i-PRF exhibits superior biological properties, including the ability of osteoblasts to proliferate, differentiate, and produce mineralized nodules more efficiently compared to other platelet-rich plasma formulations [27].

Bone regeneration is a complex, multifactorial process involving osteogenic differentiation, angiogenesis, and extracellular matrix remodeling [28]. As previously discussed, i-PRF has emerged as a promising biologically active scaffold capable of enhancing bone healing through its unique fibrin network and sustained release of bioactive factors. Unlike traditional PRP, which releases growth factors rapidly, i-PRF’s fibrin matrix prolongs the bioavailability of key osteoinductive molecules, including PDGF, VEGF, TGF-β, and bone morphogenetic proteins [4]. These molecules collectively stimulate osteoblast proliferation, extracellular matrix deposition, and neovascularization, which are essential for robust bone regeneration [4].

At the molecular level, i-PRF has been shown to upregulate RUNX2 and Osterix, two essential transcription factors for osteoblast differentiation and bone matrix mineralization [29]. Additionally, i-PRF enhances Wnt/β-catenin signaling, a well-established pathway in bone formation that regulates osteoblast activity and skeletal development [30]. The activation of the hypoxia-inducible factor-1 alpha (HIF-1α) pathway further promotes angiogenesis within the bone microenvironment, facilitating the delivery of nutrients and oxygen to regenerating tissues [31].

When compared to other orthobiologic treatments such as PRP and bone marrow aspirate concentrate (BMAC), i-PRF offers distinct advantages in terms of ease of preparation, cost-effectiveness, and sustained growth factor release. While BMAC contains mesenchymal stem cells [32], the regenerative benefits of BMAC rely heavily on cell viability and proliferation capacity, which decline with patient age [33]. Conversely, i-PRF functions as a bioactive matrix, providing a continuous release of growth factors independent of cellular content [34]. A comparative analysis of these biologics is summarized in Table 1.

The application of i-PRF in combination with bone grafts and other bone augmentation techniques has shown significant beneficial effects. i-PRF acts as a potentiator of bone regeneration by stimulating the proliferation and differentiation of osteogenic cells, promoting revascularization, and accelerating the graft consolidation process [35].

Clinical studies have also demonstrated the potential of i-PRF in bone regeneration for various applications, such as maxillary sinus augmentation procedures, regeneration of post-extraction bone defects, and fracture repair [5]. In these studies, i-PRF has been shown to accelerate bone formation, increase the density and quality of newly formed bone, and improve the integration of bone grafts. Further studies are warranted to explore optimal centrifugation protocols, platelet concentrations, and combination therapies to maximize the regenerative potential of i-PRF in orthopedic settings.

### 2.4. i-PRF in Cartilage Regeneration for Osteoarthritis

i-PRF significantly promotes chondrocyte proliferation and the mRNA levels of Sox9, type II collagen, and aggrecan when compared to PRP and the control group [36]. This beneficial effect of i-PRF on cartilage regeneration is attributed to its ability to induce the differentiation of mesenchymal stem cells into mature chondrocytes [37].

Furthermore, i-PRF has demonstrated a significant reduction in the levels of pro-inflammatory cytokines, such as interleukin-1β and tumor necrosis factor-α, in experimental models of osteoarthritis. Studies on knee osteoarthritis treatment indicate that i-PRF can reduce pain and improve joint functionality in patients, possibly due to its anti-inflammatory properties [38,39]. The improvement in pain and joint function observed in patients with knee osteoarthritis treated with i-PRF suggests that this product may be an effective therapeutic alternative for this clinical condition.

Thus, i-PRF emerges as a promising therapeutic alternative for the treatment of osteoarthritis, as it not only significantly stimulates cartilage regeneration but also exhibits potent anti-inflammatory effects that may contribute to pain reduction and substantial improvement in joint function [40]. This biological and clinical characteristic of i-PRF aligns it with the therapeutic effects and mechanisms of action of bone marrow aspirate, for example, but with less complexity in terms of extraction and application. This makes i-PRF a more accessible and convenient tool for physicians and patients with musculoskeletal disorders.

While the ability of i-PRF to promote chondrocyte proliferation and reduce inflammation is well established, recent findings suggest that its regenerative effects in cartilage regeneration extend beyond cytokine modulation and matrix protein expression. i-PRF serves as a biologically active scaffold that enhances the recruitment and retention of mesenchymal stem cells at cartilage injury sites, providing a localized microenvironment rich in bioactive factors that drive chondrogenesis [41]. At the molecular level, i-PRF has been shown to activate TGF-β and Wnt/β-catenin signaling pathways, both of which are critical for chondrogenic differentiation and extracellular matrix formation [42]. Additionally, i-PRF appears to increase expression of IGF-1 and bone morphogenetic protein 2, which play key roles in cartilage repair by stimulating proteoglycan and collagen type II synthesis while suppressing the catabolic enzymes that contribute to cartilage degradation [43,44,45,46].

Although PRP has been widely explored as an intra-articular therapy for osteoarthritis, its short half-life and rapid growth factor release may limit its long-term effectiveness [4,47]. In contrast, i-PRF’s fibrin network provides sustained growth factor delivery, allowing for prolonged biological activity in the joint space [48]. Moreover, when compared to hyaluronic acid, which primarily functions as a joint lubricant, i-PRF offers both anti-inflammatory and regenerative benefits, making it a more versatile therapeutic option [4]. Interestingly, while bone marrow aspirate concentrate remains one of the most commonly used biologic treatments for osteoarthritis, it requires invasive extraction techniques and variable stem cell counts, depending on patient age and health status [49]. i-PRF, in contrast, can be obtained with a simpler, minimally invasive preparation, making it a more accessible and cost-effective alternative for cartilage regeneration [4].

## 3. Clinical Applications

The use of platelet-rich plasma in the treatment of knee osteoarthritis and other joint conditions already has a solid foundation in clinical research and high-impact publications [50]. Several studies have demonstrated improvements in pain and joint function with the application of platelet-rich hemoderivatives in musculoskeletal conditions, particularly with the use of PRP. Regarding i-PRF, however, a preclinical study evaluated its effect on cultured chondrocytes and osteochondral regeneration in critical-sized osteochondral defects in rabbit knees, compared to PRP [51]. The results demonstrated that i-PRF significantly increased chondrocyte proliferation and the gene expression levels of chondrogenic markers, such as Sox9, type II collagen, and aggrecan, compared to PRP. Additionally, the use of i-PRF showed a significant reduction in the levels of pro-inflammatory cytokines, such as IL-1β and TNF-α, in experimental models of osteoarthritis. Regarding bone regeneration, the results indicated that both i-PRF and PRP were capable of promoting bone formation, with i-PRF showing superior performance in inducing osteochondral tissue regeneration [51].

Studies on i-PRF are still in their early stages, but the initial results are encouraging and highly promising. Several studies have successfully used solid PRF for cartilage repair [52]. Kemmochi et al. developed a method for using PRF in the repair of meniscal injuries [53]. They observed significant clinical and radiological improvement with the use of PRF compared to the control. Other researchers suggest that combining i-PRF with hyaluronic acid or stem cell therapies may provide synergistic benefits, enhancing both cartilage regeneration and joint lubrication [54]. Future studies are exploring the use of i-PRF as a carrier for bioactive molecules [48,55], potentially allowing for the controlled release of chondrogenic and anti-inflammatory agents directly at the injury site. As research continues to optimize the centrifugation parameters and injection protocols for i-PRF, its role in cartilage tissue engineering and regenerative medicine is expected to expand, offering a minimally invasive, cost-effective, and biologically active solution for osteoarthritis and other degenerative joint disorders.

## 4. Cost-Effectiveness of i-PRF

i-PRF is an interesting alternative in terms of cost-effectiveness when compared to other platelet-rich plasma formulations [3,56]. The single centrifugation process, the ability to obtain a sufficient amount of product from a small blood sample, the reduced preparation time, and the simplicity of i-PRF preparation contribute to its potential to lower costs compared to other complex and expensive biological therapies. Although the initial costs of centrifuge equipment and the necessary materials for i-PRF preparation may require an upfront investment, the medium-term use of i-PRF can be economically advantageous compared to other therapeutic options. In addition to its low cost, the rapid preparation of i-PRF, the simplicity of the procedure, and the high availability of the patient’s own blood sample further enhance its cost-effectiveness compared to other biological therapies [57,58].

In clinical practice, these characteristics make i-PRF one of the most viable and accessible biological products for patients and physicians, particularly in public healthcare settings and systems with limited resources. Such advantages facilitate the broader adoption of the method and the application of i-PRF, benefiting a larger number of patients with musculoskeletal and osteoarticular conditions. Ultimately, the favorable cost-effectiveness profile of i-PRF has the potential to expand the use of regenerative medicine.

## 5. Future Directions

Over the years, modifications to standard PRF preparation have led to next-generation formulations, including Advanced PRF (A-PRF), Titanium PRF (T-PRF), and Concentrated PRF (C-PRF). These formulations differ from traditional PRF and i-PRF in their centrifugation protocols, fibrin architecture, and growth factor retention capabilities, resulting in variations in biological properties and potential clinical applications.

A-PRF (Advanced PRF): A-PRF is produced using lower centrifugation forces and extended centrifugation times, resulting in a more porous and flexible fibrin network that retains higher numbers of leukocytes and platelets. This increased leukocyte concentration enhances growth factor release over an extended period, making A-PRF particularly useful in applications requiring sustained regenerative activity, such as bone grafting, non-union fractures, and soft tissue healing [59,60].

T-PRF (Titanium PRF): Unlike standard PRF, which is prepared using glass or plastic tubes, T-PRF is obtained using titanium-coated tubes. Research suggests that titanium enhances fibrin polymerization, resulting in a denser, more structured fibrin matrix. This may improve cell adhesion and retention, making T-PRF a promising alternative for bone defect healing and guided tissue regeneration [61].

C-PRF (Concentrated PRF): C-PRF is generated by removing excess plasma from the PRF clot, leading to a denser fibrin structure with higher concentrations of growth factors. Due to its prolonged bioactivity, C-PRF has been explored for chronic wound healing, cartilage repair, and long-term tissue regeneration strategies [62].

These next-generation PRF formulations offer distinct advantages over standard PRF and i-PRF, particularly in situations where longer-term growth factor release and structural support are required. While i-PRF remains the only fully injectable formulation, these solid PRF derivatives may complement i-PRF in regenerative orthopedic procedures where a three-dimensional fibrin scaffold is beneficial.

## 6. Conclusions

The future prospects for the use of i-PRF in orthopedics are promising, particularly concerning its potential for regenerating musculoskeletal and joint tissues. There are already different generations of PRF, such as A-PRF, T-PRF, and C-PRF. Each of these peripheral blood-derived products has unique physiological characteristics and growth factor release profiles, making them viable options not only in dentistry but also in other medical fields.

These next-generation options, considered evolutions of i-PRF, tend to behave differently from the standard product, offering new regenerative possibilities. A-PRF enhances leukocyte retention and sustained growth factor release, T-PRF utilizes titanium-induced fibrin polymerization for improved cell attachment, and C-PRF offers higher growth factor concentrations for prolonged bioactivity. While i-PRF remains the only fully injectable PRF formulation, these solid PRF derivatives may provide complementary regenerative benefits in certain orthopedic applications. All these next-generation products benefit from simplified preparation, high availability of blood samples, and low cost, with potential advantages in platelet concentration and increased release of cytokines and growth factors relevant to orthopedic applications. Thus, it is expected that the refinement of peripheral blood concentrate formulations will further expand the use of regenerative medicine in orthopedics, providing increasingly effective, accessible, and tailored therapies for various conditions.

In summary, i-PRF appears to be an attractive option for biological therapy in terms of cost-effectiveness, owing to its low production cost and simplicity of collection and application, making it more accessible to patients compared to other more complex and expensive options.

## Figures and Tables

**Figure 1 cimb-47-00239-f001:**
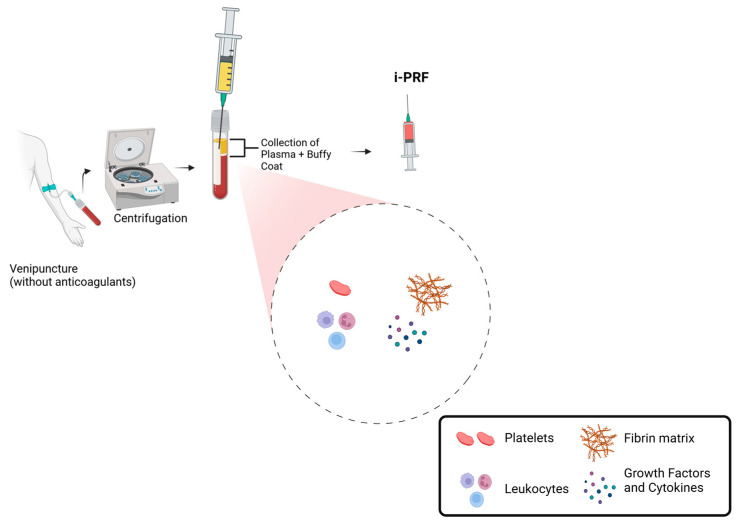
Schematic representation of i-PRF preparation and composition.

**Table 1 cimb-47-00239-t001:** Comparative overview of i-PRF, PRP, and BMAC.

Feature	i-PRF	PRP	BMAC
Preparation Method	Venipuncture and Single low-speed centrifugation	Venipuncture and centrifugation	Bone marrow aspiration (requires Jamshidi needle) and centrifugation
Anticoagulant Use	No	Yes	Yes
Fibrin Matrix Formation	Flexible and rich fibrin matrix (injectable)	No fibrin matrix	No fibrin matrix
Cellular Content	Platelets, leukocytes, cytokines, growth factors	Platelets, leukocytes, cytokines, growth factors	Mesenchymal stem cells, platelets, growth factors, cytokines
Growth Factor Release Duration	Sustained (up to 14 days)	Rapid (minutes to several hours)	Varies (depending on MSC activity)
Inflammatory Modulation	Strong anti-inflammatory effect	Moderate anti-inflammatory effect	Anti-inflammatory and regenerative effects
Clinical Applications	Osteoarthritis, cartilage repair, soft tissue healing	Osteoarthritis, tendon injuries, sports medicine	Non-union fractures, bone regeneration, cartilage defects
Cost and Invasiveness	Low cost, minimally invasive (blood draw)	Moderate cost, minimally invasive (blood draw)	High cost, invasive (bone marrow extraction)

## Data Availability

No new data generated.

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
