# Peer review of "The Role of Injectable Platelet-Rich Fibrin in Orthopedics: Where Do We Stand?"

_cimb, 2025, doi:10.3390/cimb47040239_

Round 1
Reviewer 1 Report
Comments and Suggestions for Authors
The review, "The Role of Injectable Platelet-Rich Fibrin in Orthopedics: Where Do We Stand?". Is a friendly guide that will help readers to understand the main aspects of i-PRF. It's a quick and easy read that summarizes the main findings of this platelet-rich fibrin. It briefly mentions the differences in the methodology used to obtain this injectable fibrin-rich platelet derivative vs the conventional procedure for obtaining PRF. It also pointed out the technical and biological differences between them. The paper provide by sections the main biological properties and applications in the medical field. This includes uses of I-RF in: i) wound healing; ii) antibacterial; iii) drug carrier; iv) bone regeneration; v) cartilage regeneration, and vi) in osteoarthritis. Finally, manuscript abord the clinical applications and its ability to modulate different molecular effectors in different pathologies. The authors also briefly discuss cost effectiveness.
I really think it's a great review, but when I was looking at other papers on this topic, I noticed that there are quite a few very similar ones. One of them is a quite new revision of this topic, "Ten years of injectable platelet-rich fibrin" by Richard J. Miron et al, 2024. In this review the authors describe in detail the methods used to obtain i-PRF, then they talk about how it can be used in regenerative dentistry, implantology, endodontics, orthopedics, sports medicine, treating diabetic ulcers, facial aesthetics, and even hair regeneration. This review also provides a lot of evidence from studies of the histological, serological, biochemical, and molecular effects of i-PRF compared to PRF. They also talk about how the method can be improved for the isolation and concentration of i-PRF, and how it could be used in the future to release biomolecules in a controlled way.
The paper is mentioned by the authors of the present proposal, although the paper is cited in a non-core part of the paper. Please could the authors answer the following questions and comments to identify the novelty of the present making reference to the previous reviews such as the one cited
Although the title of the proposal is specifically on Orthopedics vs a general title of Miron et al 2024, the thematic content and structure of both works is quite similar.
Could you indicate substantive differences between the two papers?
What is the novel part of the proposal?
If the focus is on orthopedics, I would suggest focusing on and breaking down points 2.3 and 2.4, which are the ones that essentially focus on the orthopedics part throughout the paper. I would kindly ask the authors to indicate the substantive differences in these areas vs. Miron's paper.
If there is not important differences, I believe that the title is not appropriate, as it is more of a general review of the properties and applications of i-PRF in the field of medicine, as it is structured in Miron's paper.
Could you indicate, in the context of the readers of Curr. Issues Mol. Biol., what is the main contribution in the field that could differentiate this review from the present proposal?
Author Response
Dear Reviewer,
Thank you for your thoughtful feedback and for recognizing the value of our review. We appreciate your insights regarding the similarities with Miron et al. (2024) and your request to clarify the novelty of our work. Below, we address your concerns point by point.
- Substantive Differences Between This Review and Miron et al. (2024):
While Miron et al. (2024) provide a broad and detailed review of injectable platelet-rich fibrin (i-PRF) across multiple medical disciplines, our manuscript is specifically focused on orthopedic applications. Unlike Miron’s paper, which discusses fields such as dentistry, implantology, and facial aesthetics, our review is dedicated to the role of i-PRF in orthopedic tissue regeneration, particularly bone and cartilage repair, osteoarthritis treatment, and musculoskeletal applications.
Additionally, our review provides:
- An in-depth discussion of orthopedic molecular mechanisms, highlighting i-PRF’s interaction with bone and cartilage cells, inflammatory pathways, and its regenerative potential in joint tissues.
- A focused comparison between i-PRF and other orthopedic orthobiologics, such as PRP and BMAC, emphasizing its clinical relevance in orthopedic practice.
- A detailed analysis of orthopedic-specific clinical evidence, which is not extensively covered in Miron’s review.
- Novel Contributions of This Review:
- Our work presents an updated synthesis of i-PRF’s role specifically in orthopedic regenerative medicine, summarizing recent findings and ongoing clinical applications.
- We analyze the clinical utility of i-PRF in musculoskeletal injuries, fractures, and osteoarthritis, emphasizing its translational potential.
- The review highlights gaps in orthopedic research regarding i-PRF and proposes future directions for its application in orthopedic tissue engineering and regenerative medicine.
- Expanding the Orthopedic Focus (Sections 2.3 and 2.4):
Based on your recommendation, we will further expand the discussion on bone and cartilage regeneration by:
- Providing additional molecular insights into i-PRF’s role in osteogenesis and chondrogenesis.
- Comparing i-PRF with conventional orthopedic biomaterials (e.g., PRP, BMAC) to reinforce its distinct advantages.
- Incorporating more orthopedic-focused clinical data to differentiate our review from Miron et al.
- Title Revision:
We understand your concern that the title may suggest a broader scope. However, as our primary focus is on orthopedic applications, we believe the current title remains appropriate. That said, we are open to modifying it if the editorial team suggests a more specific phrasing. - Relevance to CIMB Readers:
Given CIMB’s focus on molecular biology, we will ensure our revised manuscript strengthens discussions on:
- Molecular pathways influenced by i-PRF in bone and cartilage regeneration.
- Key signaling molecules (cytokines, growth factors, extracellular matrix interactions) involved in orthopedic healing.
- The potential of i-PRF as a biologically active scaffold for drug delivery in orthopedic tissues.
We appreciate the opportunity to clarify these points and will revise the manuscript accordingly. Thank you again for your valuable comments, and we look forward to your further feedback.
Reviewer 2 Report
Comments and Suggestions for Authors
In this review article, Ramos Costa et.al. have provided a succinct but comprehensive overview of the topic of injectable platelet-rich fibrin. They have summarized the simple, cost-effective process of producing this autologous platelet concentrate from blood samples, its anti-inflammatory and regenerative effects, and its anti-bacterial properties. They have also discussed its potential use for sustained drug delivery for small and large molecules, and its potential applications in the areas of bone regeneration and osteoarthritis treatments. In each of these areas, they have summarized recent findings and have provided literature references supporting these.
The article is well written and is of interest to a broad audience. I have one comment/suggestion for the authors. In the Conclusions section (lines 244 to 249, page 6), the authors mention “There are already different generations of PRF, such as A-PRF, T-PRF, and C-PRF. Each of these peripheral blood-derived products has unique physiological characteristics and growth factor release profiles, making them viable options not only in dentistry but also in other medical fields. These options, considered evolutions of i-PRF, tend to behave differently from the standard product, offering new regenerative possibilities.”. Please elaborate more on what exactly are A-PRF, T-PRF, and C-PRF, and how these products differ from i-PRF. Please also provide more details on how they behave differently from i-PRF and what are the new regenerative possibilities are provided by these that are not possible with i-PRF.
Author Response
Dear Reviewer,
Thank you for your valuable suggestion regarding A-PRF, T-PRF, and C-PRF. Based on your feedback, we have:
Created a new section (Section 5: Future Directions) dedicated to next-generation PRF formulations, providing a detailed discussion on how they differ from i-PRF and their potential regenerative applications.
Condensed the reference to these PRF types in the Conclusion, directing readers to Section 5 for further details.
We appreciate your insightful recommendation and believe this revision enhances the manuscript’s clarity and structure, ensuring that next-gen PRF advancements are highlighted in the proper context.
Round 2
Reviewer 1 Report
Comments and Suggestions for Authors
The new version provided by the authors makes a better distinction with the Miron 2024 review. I agree with the authors that while the article by Miron et al. (2024) provides a comprehensive and detailed review of injectable platelet-rich fibrin (i-PRF) in multiple medical disciplines, such as dentistry, implantology and facial aesthetics, the section dedicated to applications in bone regeneration was in general, similar to the first version of the review proposed by the authors.
The extensive information included in the second version provides the reader with a compilation of greater, more detailed and specific knowledge about the role of i-PRF in bone regeneration, as well as a comparison with other presentations of PRF. The authors have included precise and abundant information, that allows us to differentiate and contribute more knowledge to what was reported by Miron 2024 in the section on bone regeneration.
The inclusion of Table I also allows the reader to clearly see the differences between i-PRF with PRP and BMAC, emphasizing its clinical relevance, particularly in Orthopedic practice.
The title of the manuscript sounds appropriate, after the extensive information included in this new version.
The new version of the review already includes an extensive discussion of the probable molecular pathways influenced by i-PRF, although it is clear that some of these pathways remain to be experimentally demonstrated.
It was also a good idea to include Section 5 which outlines the areas of opportunity and the knowledge that remain to be determined regarding the use of i-PRF.
Finally, I would like to thank the authors for their efforts in responding to my suggestions. Now, in this new version, the specific contribution of the paper in Orthopedics is clearer, making now suitable for publication in CIMB.